# Online Assessment of Motor, Cognitive, and Communicative Achievements in 4-Month-Old Infants

**DOI:** 10.3390/children9030424

**Published:** 2022-03-16

**Authors:** Corinna Gasparini, Barbara Caravale, Valentina Focaroli, Melania Paoletti, Giulia Pecora, Francesca Bellagamba, Flavia Chiarotti, Serena Gastaldi, Elsa Addessi

**Affiliations:** 1Department of Dynamic and Clinical Psychology and Health Studies, Sapienza University of Rome, Via Degli Apuli 1, 00185 Rome, Italy; cori.gaspar@gmail.com (C.G.); melania.paoletti@gmail.com (M.P.); francesca.bellagamba@uniroma1.it (F.B.); 2Department of Developmental and Social Psychology, Sapienza University of Rome, Via Dei Marsi 78, 00185 Rome, Italy; barbara.caravale@uniroma1.it; 3Unità di Primatologia Cognitiva e Centro Primati, Istituto di Scienze e Tecnologie della Cognizione, Consiglio Nazionale delle Ricerche (CNR), Via Ulisse Aldrovandi, 16/b, 00197 Rome, Italy; valentina.focaroli@gmail.com (V.F.); giuliapecora.psi@gmail.com (G.P.); serena.gastaldi@istc.cnr.it (S.G.); 4Center for Behavioral Sciences and Mental Health, Istituto Superiore di Sanità, Viale Regina Elena, 299, 00161 Rome, Italy; flavia.chiarotti@iss.it

**Keywords:** infants, online assessment, development, Bayley Scales, COVID-19

## Abstract

Remote methods for data collection allow us to quickly collect large amounts of data, offering several advantages as compared to in-lab administration. We investigated the applicability of an online assessment of motor, cognitive, and communicative development in 4-month-old infants based on several items of the Bayley Scales of Infant Development, 3rd edition (BSID-III). We chose a subset of items which were representative of the typical developmental achievements at 4 months of age and that we could administer online with the help of the infant’s caregiver using materials which were easily available at home. Results showed that, in a sample of infants tested live (N = 18), the raw scores of the BSID-III were significantly correlated with the raw scores of a subset of items corresponding to those administered to a sample of infants tested online (N = 53). Moreover, for the “online” participants, the raw scores of the online assessment did not significantly differ from the corresponding scores of the “live” participants. These findings suggest that the online assessment was to some extent comparable to the live administration of the same items, thus representing a viable opportunity to remotely evaluate infant development when in-person assessment is not possible.

## 1. Introduction

Online testing during videoconference has represented a unique opportunity to continue developmental studies and clinical practice during the COVID-19 emergency. It has allowed us to overcome the social restrictions imposed by the pandemic, and to enhance accessibility and support for families [1,2,3]. The use of technology in healthcare, including synchronous video consultation, reduces costs and improves access to health services [4], and these positive effects may well extend beyond the pandemic emergency. 

The use of online methodologies is also becoming a common practice in psychological research to obtain large amounts of data [5,6,7,8] and to reach people who might not be able to participate in laboratory settings [9]. Online methodologies are flexible and convenient to implement, as they allow fast data collection [10], increase sample diversity, and facilitate longitudinal research [11] (all features which are not always easily achievable with in-person testing). In addition, researchers are able to observe children in a familiar, more naturalistic environment. Nonetheless, online data collection can be also challenging: low-income families could be disadvantaged, likely having a lower availability of technological devices at home, and the safeguarding of privacy and data security is required. Moreover, when online assessments involve young children aged less than two years it can be difficult to ensure the standardization of the procedures, especially when the tasks are administered by the child’s caregiver [12], and to clearly observe the child’s non-verbal reactions to the items. 

Over the last two years, there has been a flourishing of online developmental studies investigating a broad range of topics [13]. The findings obtained through remote testing of preschool children have generally been promising: 1- to 4-year-old children’s reaction-time data in a word-recognition paradigm obtained through an online assessment compared favorably with eye tracking and an in-person storybook approach [14]. Similarly, 3- and 4-year-old children’s performances in the change-of-location false belief task were similar between in-lab and online test settings [15]. Moreover, 4- to 5-year-old children tested on a battery of eight standardized cognitive tasks (involving verbal comprehension, fluid reasoning, visual spatial and working memory, attention and executive functioning, social perception, and numerical skills) performed similarly for most of these tasks regardless of whether they were tested live or online [16]. Additionally, online methods proved to be particularly useful when recording mealtimes of children younger than three years of age, allowing to better preserve the ecological validity of the observations [17]. 

So far, the number of reports on remote testing in infants is more limited as compared to older children and most of these studies involved the use of a looking-time methodology, e.g., [18]. However, results are not always consistent with live testing, probably because of methodological constraints on the stimuli presentation [19,20,21]. To our knowledge, only a few studies have attempted to perform online motor development assessments on infants, by remotely evaluating reaching and sitting skills in 3-to-5-month-old infants [22] and assessing the feasibility for parents of performing home video recordings of their infant to be used for offline motor development assessments [23]. However, there is still little evidence of structured and standardized online neurodevelopmental assessments to be used with infants.

In the present study, we aimed to test the extent to which a subset of items of the Bayley Scales of Infant and Toddler Development, 3rd edition (BSID-III) [24], could be administered online to obtain information about 4-month-old infants’ motor, cognitive and communicative skills, to be used for research purposes when in-person administration of the full scale is not feasible. The BSID-III is one of the most widely used tools for assessing the neurodevelopment of children aged zero to three years and is considered highly informative in terms of motor, cognitive, and communicative abilities. As other developmental assessment tools, the BSID-III is typically administered face-to-face by a qualified examiner in the presence of a caregiver. Specifically, we assessed whether (i) the subset of BSID-III items chosen for the online administration was representative of the full BSID-III scale by evaluating if, in a sample of infants tested live, the raw scores of the subset of BSID-III items were significantly correlated with the raw scores of the full BSID-III scale. Moreover, we assessed whether (ii) the online administration of a subset of BSID-III items was comparable to the live administration of the same BSID-III items by evaluating if the raw scores of the subset of BSID-III items chosen for the online administration differed between the sample of infants tested online and the sample of infants tested live.

## 2. Materials and Methods

### 2.1. Participants

We tested a total of 71 full-term Italian infants at 4 months of age. Fifty-three infants were observed online (“online participants”, gestational age: M = 39.51 weeks, SD = 1.1; age: M = 4.11 months, SD = 0.217) and 18 infants were administered the BSID-III [24] at home (“live participants”, gestational age: M = 39.60 weeks, SD = 1.13, age: M = 4.19 months, SD = 0.15). Each infant was tested once, either online or live. 

Both “online” and “live” participants were recruited during mother’s pregnancy or soon after birth for a larger longitudinal research project on the development of infant feeding behavior. The recruitment was made through advertising via social media, posters in pediatricians’ offices, and the newsletter of the magazine “Uppa Magazine” addressed to parents. Children who were born before 37 weeks of gestation, with congenital abnormalities, severe neurological deficits, from twin births, and/or systematically exposed to a second language other than Italian, were excluded from the sample. Parents, who accepted to participate with their infant provided a written informed consent for taking part in the study and to be video recorded. Figure 1 shows a flow chart of infant recruitment. All procedures were approved by the Ethics board of the Department of Dynamic and Clinical Psychology and Health Studies of Sapienza University of Rome (Prot. n. 0000315, 14 April 2020 and n. 0001209, 15 December 2020). All data were collected during a period of about seven months. 

### 2.2. Measures

We developed an online assessment of infant motor, cognitive, and communicative development based on a subset of items of the BSID-III [24]. BSID-III provides a norm-referenced developmental index for each of the following domains: cognitive (COG), receptive language (RL), expressive language (EL), fine motor (FM), and gross motor (GM). For our online assessment, for each subscale we selected a subset of items on the basis of three criteria: (i) their representativeness of the typical developmental abilities at 4 months of age (choosing, as the first item of each subscale, the item corresponding to the starting point of the BSID-III based on the age range, i.e., D), (ii) the availability at the infant’s home of materials as similar as possible to those used in the BSID-III, and (iii) the ease for the caregiver to administer the item and take care of the child at the same time. The final list of items administered online included, for each subscale: COG items 7, 8, 10, 14, 16, 17, 18, 21; RL items 3, 4, 5, 6, 8; EL items 1, 2, 3, 4, 5, 6; FM items 5, 6, 10, 11, 12, 13, 14; GM items 9, 10, 11, 12, 13, 14, 15, 16, 17, 18, 19, 21. 

Only the sample of N = 53 “online” participants received the online assessment, carried out by an experienced examiner (as reported in the “Procedure” section). For each item, we scored whether the infant showed (1) or not showed (0) a specific ability, based on the BSID-III manual. For some of the 53 “online” participants we could not include a few items in the calculation of the raw score due to audio or video problems, issues with internet connectivity, or—in a very few cases—examiner’s errors. For each subscale we included in the data analyses only those participants for whom we could score all the items administered: N = 46 for the COG subscale, N = 27 for the RL subscale, N = 46 for the EL subscale, N = 46 for the FM subscale, and N = 30 for the GM subscale. Most of the missing data for the RL and GM subscales concerned items which were sometimes difficult to observe or administer by the caregiver (e.g., RL6 “Turns to the source of the sound at least one time” and GM13 “Keeps the head in balance”). 

For the “live” participants (N = 18) the standard BSID-III was administered at home by a trained examiner during a single home visit. In addition to the total score usually computed for each subscale following BSID-III manual rules, we also calculated raw scores considering only the subset of items chosen for the remote administration to the “online” participants (i.e., summing items scored as “1” for each subscale). 

All mothers (N = 71) also completed a socio demographic questionnaire. 

### 2.3. Procedure

As reported in the “Measures” section, all child assessments were conducted by a qualified professional with a degree in developmental psychology and trained to administer the BSID-III scale. Specifically, for the sample of “live participants”, BSID-III was administered at the child’s home by the examiner through observation and direct interaction with the child, whereas the sample of “online” participants received the assessment remotely through observations carried out via video conference using Skype or Jitsi Meet and recorded through the software Open Broadcaster Software (OBS) Studio for subsequent coding. A second observer scored 20% of the videoclips; the index of concordance was 1 for all items, except COG21, RL4, RL8, EL6 (0.91), RL6, EL2, FM11, FM13, GM17, GM18 (0.82). Each online assessment was carried out during a single video conference session while the child was at home, using the family’s own camera-equipped computer, phone, or tablet, and lasted approximately 25–30 min. To minimize interference, caregivers were previously instructed by the experimenter about the materials to put at the infant’s disposal (including a book with big and colored figures, a ring, a rattle or a noisy object, and a small toy), and were asked to remove not necessary toys and to be alone with the infant during the assessment. During the online assessment, parents were asked to sit on the floor next to the infant and to provide several infant’s postural adjustments according to the examiner’s instructions. The examiner continuously guided the caregiver in administering the items to the child and in introducing specific items during the session. 

### 2.4. Data Analyses

First, for the “live” participants, for each subscale we performed Spearman correlations between the raw score of the full BSID-III and the raw score of the subset of items corresponding to those administered remotely to the “online” participants. Second, we compared “online” participants and “live” participants regarding demographic characteristics and raw scores of the subset of the BSID-III items administered remotely (“online” participants) or corresponding to those administered remotely to the “online” participants (“live” participants). We used the Chi-squared test to assess differences for categorical variables. For quantitative variables, we previously tested the heterogeneity of variances by the Levene test; we then assessed differences between group by means of the Student’s t-test, or the Welch t-test in case of heterogeneous variances. 

## 3. Results

Among “live” participants, for each subscale the raw score of the full BSID-III was significantly correlated with the raw score obtained by summing the scores of the items corresponding to the subset of BSID-III items administered remotely to the “online” participants (cognitive: r_s_ = 0.549, *p* = 0.018, receptive language: r*_s_* = 0.609, *p* = 0.007, expressive language: r*_s_* = 0.839, *p* < 0.001, fine motor: r*_s_* = 0.940, *p* < 0.001, gross motor: r*_s_* = 0.984, *p* < 0.001). This suggest the subset of BSID-III items chosen for the online administration was representative of the full BSID-III scale administered to the participants tested in presence. 

As reported in Table 1, “online” participants and “live” participants did not significantly differ for demographic characteristics as age, gender, number of siblings in the household, birthweight for gestational age, level of maternal education, or income. Moreover, as reported in Table 2, “online” participants and “live” participants did not significantly differ for the raw scores of the subset of BSID-III items administered remotely to the “online” participants (which were extrapolated from the full BSID-III scale for the “live” participants). This suggests that the online administration of a subset of BSID-III items was comparable to the live administration of the same BSID-III items. 

## 4. Discussion

In this study, we aimed to assess whether (i) a subset of BSID-III items chosen to be administered online was representative of the full BSID-III scale and (ii) the online assessment of this subset of BSID-III items was comparable to the live administration of the same BSID-III items. For “live” participants (who were administered the full BSID-III at home) we found significant correlations between the raw scores of each BSID-III subscale and the raw scores of a subset of BSID-III items corresponding to those administered remotely to a group of “online” participants. Moreover, we showed that the raw scores obtained for each subscale in the “online” sample were not significantly different from the corresponding scores computed for the “live” sample (as well as socio-demographic characteristics known to potentially affect the BSID-III scores). This suggests that the BSID-III items administered during the online assessment are somewhat representative of the full subscales of the BSID-III, at least for providing a qualitative assessment of child development and that the live and online administration of this subset of BSID-III items were to some extent comparable. 

Our study represents the first attempt to remotely assess infants’ neurodevelopment in multiple domains. In fact, previous online studies involving infants and young children mainly collected data about single abilities (such as visual preference, behavioral responses to animations, performance in cognitive tasks or motor development, e.g., 14–16, 22). Strengths of our methodology included the possibility of collecting reasonably large observational datasets regarding the very early phases of motor, cognitive, and communicative development in a relatively short period of time, as also observed in other studies using online methods to evaluate different aspects of children’s development, e.g., [25]. Additionally, remotely observing the child at home (rather than in a laboratory or clinical settings) may be advantageous in terms of preserving the ecological validity of the data when evaluating some of the abilities targeted in the present study, such as early spontaneous communication [26]. Similarly, online assessments also provide the advantage that parents and examiners can easily postpone their meeting, if needed, and reschedule it when the infant is in a quiet alert state, which is fundamental for an optimal performance in perceptual, cognitive, and motor domains [27].

Similarly to other studies using online methods to assess children’s development, the present study has some limitations to consider, e.g., [13,27,28]. Although collecting data via live video connection was easily manageable for both the examiner and the families, the data quality was sometimes affected by this procedure. For instance, technological difficulties (e.g., audio or video problems) and issues with internet connectivity forced us to exclude some observations from the sample. Moreover, since the study took place at participants’ home, the environmental context differed for each participant and some interfering factors, including the furniture disposition, the lighting condition, the presence of potentially distracting objects, and the necessity to manage the setting by the caregiver (e.g., position of the digital device, child’s postural adjustments), may have influenced the quality of the observations. Furthermore, the fact that the parents directly administered the items under the examiner’s guidance may have further increased the variability of the assessment. Additionally, our online assessment may become more difficult to implement when children get older and become capable of independent locomotion, especially when one parent is alone at home with the child and needs to simultaneously administer the items and film the child.

Nonetheless, our exploratory study provides some evidence that online methodologies are a viable alternative to assess infants’ neurodevelopmental abilities by administering a subset of items drawn from the BSID-III when in-person assessment is not feasible. It should be noted that the use of this online assessment should be limited to research purposes. Indeed, a neurodevelopmental assessment carried out for clinical purposes needs a face-to-face visit, as motor, cognitive, and language subtests cannot be administered in a standardized format via telepractice. In this respect, in the recent Fourth Edition of Bayley Scales of Infant and Toddler Development (BSID-IV), the opportunity has been provided to observe children via telepractice by reviewing the BSID-IV items at the age-appropriate starting point. Still, since the Bayley Scales are not standardized in a telepractice mode, the examiner is recommended to treat these data with caution [29] as only qualitative information on children’s motor, cognitive, and communicative skills can be obtained, and standard scores cannot be computed. 

Finally, due to COVID-19 restrictions we could test only a small sample of children live and could not test the same children both live and online. This should be the goal of future studies, in order to fully evaluate the robustness of our online assessment. 

## 5. Conclusions

In the present study, we assessed the feasibility of an online assessment of motor, cognitive, and communicative development in 4-month-old infants based on several items of the Bayley Scales of Infant Development, 3rd edition, to be used for research purposes when in-person administration is not possible.

Our online assessment was to some extent comparable to the live administration of the same items, thus representing a viable opportunity to remotely evaluate infant development and to early detect information about motor, cognitive, and communicative skills. In fact (i) for live participants there were significant correlations between the raw scores of each BSID-III full subscale and the raw scores of the subset of BSID-III items corresponding to those administered remotely to a group of “online” participants, and (ii) the subset of the BSID-III items administered online did not significantly differ from the corresponding scores obtained in the sample of infants tested live. 

Thus, despite some limitations, the large amount of infant data collected during the period of social restrictions imposed by COVID-19, and the precious developmental information obtained for motor, cognitive, and communicative domains, as well as the minimal costs for implementing this procedure, highlight the value of this online methodology for future investigations, especially when performing a live developmental assessment is not possible. 

## Figures and Tables

**Figure 1 children-09-00424-f001:**
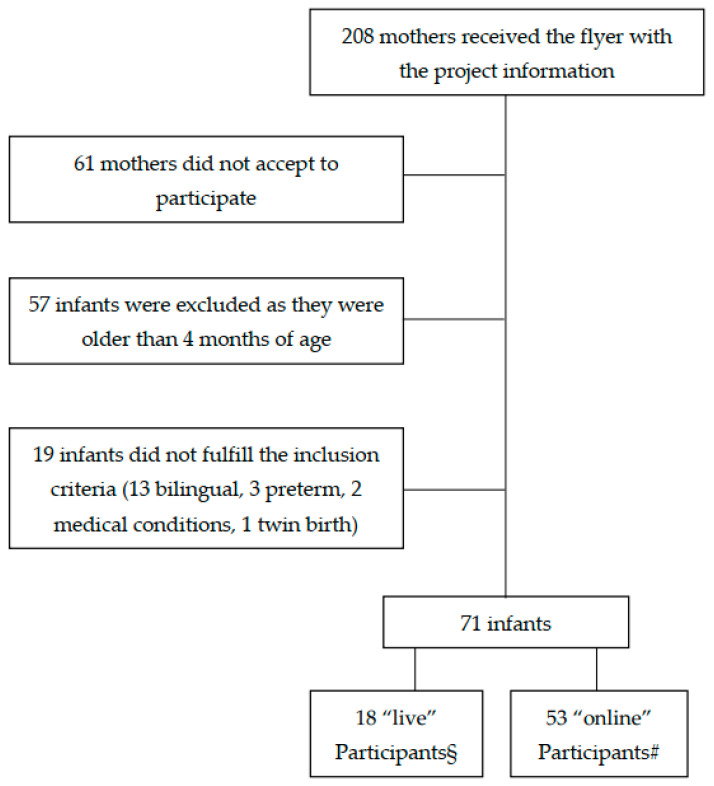
Flow chart of infant recruitment. § These participants were tested before COVID-19 restrictions were applied in Italy. # These participants were tested after the applications of COVID-19-related restrictions.

**Table 1 children-09-00424-t001:** The table reports the descriptive values for the demographic variables for each group of participants (tested “online” or “live”), and the statistical values (Student’s *t*-test or Chi square) for the comparisons between the two groups.

Variable	Group	Testing Modality		Student’s *t*-test/Chi Square
		Online (N = 53)	Live (N = 18)	
Infant age (months, mean ± standard deviation)		4.11 ± 0.21	4.19 ± 0.15	t (69) = −1.217, *p* = 0.228
Infant gender (*n* and %)	Males	27 (50.9)	10 (55.6)	χ^2^_1_ = 0.115, *p* = 0.735
	Females	26 (49.1)	8 (44.4)	
Number of siblings (*n* and %)	Zero	30 (56.6)	9 (50.0)	χ^2^_3_ = 3.154, *p* = 0.368
	One	21 (39.6)	7 (38.9)	
	Two	2 (3.8)	1 (5.6)	
	Three	0	1 (5.6)	
Infant birth weight for gestational age (*n* and %)	Adequate for gestational age	42 (79.2)	14 (77.8)	χ^2^_2_ = 2.836, *p* = 0.242
	Small for gestational age	5 (9.4)	0	
	Large for gestational age	6 (11.3)	4 (22.2)	
Maternal education (*n* and %)	Middle school	0	1 (5.6)	χ^2^_4_ = 3.348, *p* = 0.501
	High school	2 (3.8)	1 (5.6)	
	Bachelor’s	9 (17.0)	2 (11.1)	
	Master’s	33 (62.3)	11 (61.1)	
	PhD or equivalent	9 (17.0)	3 (16.7)	
Income (EURO) (*n* and %)	<EUR 10,000	1 (1.9)	0	χ^2^_9_ = 6.134, *p* = 0.726
	EUR 10,000–19,000	6 (11.3)	1 (5.6)	
	EUR 20,000–29,000	5 (9.4)	2 (11.1)	
	EUR 30,000–39,000	7 (13.2)	1 (5.6)	
	EUR 40,000–49,000	8 (15.1)	3 (16.7)	
	EUR 50,000–59,000	4 (7.5)	4 (22.2)	
	EUR 60,000–69,000	2 (3.8)	1 (5.6)	
	EUR 70,000–79,000	2 (3.8)	1 (5.6)	
	EUR 80,000–89,000	3 (5.7)	0	
	>EUR 100,000	2 (3.8)	0	
	Missing	13 (24.5)	5 (27.8)	

**Table 2 children-09-00424-t002:** The table reports, for each subscale of the online developmental assessment, the descriptive values of the raw scores of the subset of BSID-III items administered remotely to the “online” participants and of the same subset of BSID-III items administered in presence to the “live” participants, along with the statistical values for the comparisons between the two groups (Levene’s test, Student’s *t*-test, Observed and Minimum detectable Cohen’s d).

	Test Modality	N	Mean	Standard Deviation	Levene’s Test (F)	Levene’s Test (*p*)	t	df	*p*	Observed Cohen’s d	Minimum Detectable Cohen’s d #
Online Assessment Subscales											
Cognitive	Live	18	7.389	0.916	2.273	0.137	−1.377	62	0.174	0.38	0.79
	Online	46	7.674	0.668							
Receptive Communication	Live	18	4.440	0.511	0.101	0.752	1.848	43	0.072	0.56	0.87
	Online	27	4.110	0.641							
Expressive Communication	Live	18	5.670	0.485	4.354	0.041	1.458	47.853	0.151 §	0.42	0.79
	Online	46	5.430	0.750							
Fine Motor	Live	18	5.610	1.577	3.945	0.051	−1.159	62	0.251 §	0.30	0.79
	Online	46	6.070	1.340							
Gross Motor	Live	17	8.240	1.821	0.096	0.759	0.825	45	0.414	0.25	0.87
	Online	30	7.800	1.690							

# at two-tailed significance level = 0.05 and power = 0.80. § Welch test.

## Data Availability

The full dataset is available at the following link: https://osf.io/ug2qm/?view_only=6fa07776ebbd4aa491d90378f40bd252. (accessed on 8 February 2022)

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
