# Peer review of "Online Assessment of Motor, Cognitive, and Communicative Achievements in 4-Month-Old Infants"

_children, 2022, doi:10.3390/children9030424_

Round 1
Reviewer 1 Report
Thank you for giving me the opportunity to read this exciting study. The study tries to develop an online assessment tool based on BSID-III to evaluate 4-month infants’ development. While the idea is interesting and the authors put much effort into the study, there are essential questions about the validity of the study results and need to be improved with significant changes of the study design.
Major comments
- Due to the very limited sample size (18 vs. 53), is it possible that no significant difference is caused by under power design, e.g., Type II error? It would be super helpful if the authors could provide more information about the power calculation.
- According to the "Measures" section, the selected subset items from BSID-III are mainly based on the convenience of online testing. How is the validity of overall child development and subset skills measured compared to the full BSID-III items?
- Did the author receive permission from the BSID-III developers to use subset items or related copyright? If not, it will highly limit the possible application of this study (if it is well designed).
- BSID-III has the raw and adjusted scores to measure children's skill development. The author only showed the raw score in Table 2. How does the raw score compare with the adjusted score? How to understand the result and compare it to author studies using BSID-III?
Author Response
Comments and Suggestions for Authors
Thank you for giving me the opportunity to read this exciting study. The study tries to develop an online assessment tool based on BSID-III to evaluate 4-month infants’ development. While the idea is interesting and the authors put much effort into the study, there are essential questions about the validity of the study results and need to be improved with significant changes of the study design.
We thank the Reviewer for evaluating our study. Please find below detailed responses to each concern.
Major comments
1. Due to the very limited sample size (18 vs. 53), is it possible that no significant difference is caused by under power design, e.g., Type II error? It would be super helpful if the authors could provide more information about the power calculation.
We recognize that the sample size in the “live”sample is small (as also reported in the Discussion section, last paragraph). The unexpected Covid-19 pandemic emergency pushed us to promptly find an alternative method to collect these precious early developmental data, without losing lots of subjects. Thanks to the convenience of the remote administration, we were able to have a much larger sample for the online tested infants. Unfortunately, the persistence of the pandemic emergency and of the restrictions did not allow us to expand the in-person sample. However, our study was powered enough to detect clinically relevant differences between the two groups. Indeed, with our sample size the Student t test, with a two-tailed significance level alpha = 0.05 and a power 1-beta = 0.80, allowed us to identify a difference between means of “live” vs.“online” subjects of medium-large size (Cohen’s d = 0.79 for Cognitive, Expressive communication and Fine motor subscales, and d = 0.87 for Receptive communication and Gross motor subscales). The effect sizes we actually observed for the subscales were small (0.38, 0.56, 0.42, 0.30, 0.25 for Cognitive, Receptive communication, Expressive communication, Fine motor and Gross motor subscales, respectively) and likely not clinically relevant. We have now added the detectable Cohen’s d next to the actual Cohen’s d in Table 2.
2. According to the "Measures" section, the selected subset items from BSID-III are mainly based on the convenience of online testing. How is the validity of overall child development and subset skills measured compared to the full BSID-III items?
As reported in the Data Analyses section and further detailed below (please see response to query # 4), for the “live” participants, who were administered in presence the full BSID-III, for each subscale we performed Spearman correlations between their raw score of the full BSID-III and their raw score of the subset of items corresponding to those administered during the online assessment to the “online” participants. This was made to assess whether the raw scores of the subset of BSID-III items chosen for the online administration were representative, for the same subjects, of the raw scores of the full set of BSID-III items administered in presence.We found significant correlations for all subscales (as reported in the Results section, first paragraph), thus we believe that the subset of the BSID-III items administered online were representative of the full set of BSID-III which is possible to administer in presence.
Moreover, to evaluate whether the items selected for the online administration may have provided a skewed view of infant development, for the live participants, for each subscale, we also assessed the correlation between the total raw score and the raw score for two different subsets of items, respectively, each including the same number of items as the originally selected set of items (for instance, for the COG subscale, for which we originally selected 8 items for the online administration, we now correlated the raw score of the first 8 items and of the last 8 items, respectively, with the full raw score). As each infant received a different number of items depending on performance, for each subscale we selected, as the last item, the lowest number of items administered for that subscale (for instance, for the Cognitive subscale, where infants received a minimum of 19 items and a maximum of 38, we selected the item 19 as the last one for all subjects). For virtually all subscales, it emerged that either the raw scores of the first items did not significantly correlate with the total raw scores or it was not possible to calculate such correlation (because of a lack of variance in infants’ performance on the first items, which was usually at ceiling level): Cognitive: rs = .212, p = .399; Receptive Communication: no variance in the first items, Fine Motor: no variance in the first items; Gross Motor: no variance in the first items. The only exception was represented by Expressive Communication, for which the raw scores on the first items was significantly correlated with the total raw score (rs = .862, p < .001), likely because subjects’ performance on this subscale showed higher variability than on the other subscales. In contrast, the raw scores of the last items always significantly correlated with the full raw scores (Cognitive: rs = .842, p < .001; Receptive Communication: rs = .978, p < .001; Expressive Communication: rs = .998, p < .001; Fine Motor: rs = .984, p < .001; Gross Motor: rs = .994, p < .001), as also observed for the items selected for the online administration, which thus did not seem to provide a skewed view of infants’ development.
3. Did the author receive permission from the BSID-III developers to use subset items or related copyright? If not, it will highly limit the possible application of this study (if it is well designed).
We have subscribed for using the BSID-III during live administration, but we did not seek for any specific permission to administer a subset of BSID-III items remotely. Nonetheless, the recent BSID-IV provides the opportunity also to observe children via telepractice by reviewing the Bayley-IV items at the age-appropriate starting point (as reported in the Discussion section, fourth paragraph).
4. BSID-III has the raw and adjusted scores to measure children's skill development. The author only showed the raw score in Table 2. How does the raw score compare with the adjusted score? How to understand the result and compare it to author studies using BSID-III?
As the online participants could not receive the full BSID-III (for the limitations of the online administration), but only a subset of BSID-III items, we could not calculate their standardized scores. To obtain comparable scores for both “online” and “live” participants, for the “live” participants we calculated, for each subscale, also the raw scores derived from the sum of the items corresponding to the subset of Bayley items administered to the “online” participants during the video assessment (thus excluding, for “live” participants, those items which were not possible to administer remotely to the “online” participants). In Table 2, we have reported the raw scores for the subset of BSID-III items used in the online assessment with the “online” participants and for the corresponding subset of BSID-III items administered in presence to the “live” participants, which we compared by the Student t test. Please also note that all infants had the same age, 4 months ± 2 weeks, and thus they had the same age-appropriate starting point (i.e., D), and that, for “live” participants, the raw scores of each subscale were significantly correlated with the standardized score of the corresponding subscale (Cognitive: rs = .795, p < .001; Receptive Communication: rs = .975, p < .001; Expressive Communication: rs = .998, p < .001; Fine Motor: rs = .958, p < .001; Gross Motor: rs = .730, p < .001).
In addition, since the “live” participants received the full BSID-III, for them we could calculate both the raw score and the standardized score for each BSID-III subscale.As reported in the Data Analyses section and above (query # 2), to assess how the raw score of the subset of BSID-III items used in the online administration compared to the rawfull BSID-III score, for the live participants we performed, for each subscale, Spearman correlations between their raw score of the full BSID-III and their raw score of the subset of items corresponding to those administered during the online assessment to the “online” participants. We found significant correlations for all subscales (reported in the Results section, first paragraph) and, thus, we believe that the subset of the BSID-III items administered online were representative of the full set of BSID-III which is possible to administer in presence.
Our intention was not to translate and validate the BSID-III to a fully online version, nor to use this online assessment in comparison to other studies using the full BSID-III in live samples. Our aim was to propose an online assessment of infant development which may be used when in-person administration of the full BSID-III (or other in-person neurodevelopmental assessments) is not feasible. Since the Bayley Scales are not standardized in a telepractice mode, it is necessary to treat these data with caution as only qualitative information on children’s motor, cognitive and communicative skills can be obtained (as we recall in the Discussion section, fourth paragraph). Of course, in case of clinical signs emerging from the online assessment, the in-person administration of the standardized scale is certainly recommended.
Reviewer 2 Report
General comments
This paper is an accurately useful for the detection of disorders motor, cognitive and communicative in infants.
Introduction
Introduction is long, I suggest a shorter a more focused intro.
The introduction would benefit from hypotheses providing a rationale.
Materials and Methods
The Measures section is hard to read.
Overall, I would recommend shorten this section.
Add a flow diagram to population and their distribution.
Results
The tables present the data nicely but a bit more explanation of the content and implications of the data in the tables would be beneficial.
Discussion
I would like to be able to distinguish a first paragraph with a summary of what was sought and what was found in this work, a discussion of the relevant findings together with what was found in other works, an explanation of the meaning of these findings, an explanation of the implications for practice clinic and suggestions for future research.
Finally, is that there is what is the practical impact of this study?
Author Response
General comments
This paper is an accurately useful for the detection of disorders motor, cognitive and communicative in infants.
We thank the Reviewer for evaluating our study. Please find below detailed responses to each concern.
Introduction
Introduction is long, I suggest a shorter a more focused intro.
We have revised and streamlined the Introduction, in the attempt of making it more compelling and straightforward.
The introduction would benefit from hypotheses providing a rationale.
In the last paragraph of the Introduction, we have attempted to better spell out our aims and expectations.
Materials and Methods
The Measures section is hard to read.Overall, I would recommend shorten this section.
We have shortened and attempted to clarify this section.
Add a flow diagram to population and their distribution.
We have included a flow chart of infant recruitment (Figure 1).
Results
The tables present the data nicely but a bit more explanation of the content and implications of the data in the tables would be beneficial.
We have added further information both in the legends of each table and in the Results section. We hope to have satisfactorily addressed the Reviewer’s comment.
Discussion
I would like to be able to distinguish a first paragraph with a summary of what was sought and what was found in this work, a discussion of the relevant findings together with what was found in other works, an explanation of the meaning of these findings, an explanation of the implications for practice clinic and suggestions for future research.
We thank the Reviewer for this comment. We have restructured the discussion, which now reports: a summary of our aims and main findings (first paragraph), a discussion of our findings in relation to previous works on the use of online methodology to investigate developmental aspects in infants and young children, including strengths and limitations of the present study (second and third paragraphs), an explanation of our findings and their implications for clinical practice (fourth paragraph), suggestions for future research (last paragraph of the Discussion).
Finally, is that there is what is the practical impact of this study?
As also reported in the Discussion section, remote assessments of neurodevelopment may be used for research purposes when in-person assessments are not feasible (not necessarily only during a pandemic emergency). Remote assessments have the advantages of allowing data collection on large samples in a relatively limited period of time and of obtaining more ecologically valid data (as children are tested in their home environment when they are in a quiet state). Unfortunately, as reported in the Discussion, our remote assessment, based on a subset of items taken for the BSID-III, cannot completely replace the live administration of the full BSID-III scale, as the administration in a standardized format via telepractice is not possible.
Round 2
Reviewer 1 Report
I did not notice a significant improvement in the manuscript according to comments, especially on the study design and how to interpret the results.